# Structural and Thermal Characterization of Protein Isolates from Australian Lupin Varieties as Affected by Processing Conditions

**DOI:** 10.3390/foods12050908

**Published:** 2023-02-21

**Authors:** Lavaraj Devkota, Konstantina Kyriakopoulou, Robert Bergia, Sushil Dhital

**Affiliations:** 1Department of Chemical and Biological Engineering, Monash University, Clayton, VIC 3800, Australia; 2Archer-Daniels-Midland (ADM), James R. Randall Research Centre, Decatur, IL 62521, USA

**Keywords:** protein profile, secondary structure, circular dichroism, α-conglutin, β-conglutin, circular dichroism

## Abstract

Proteins from the full and defatted flours of *L. angustifolius* cv Jurien and *L. albus* cv Murringo were prepared using alkaline extraction and iso-electric precipitation. Isolates were either freeze dried or spray dried or pasteurized at 75 ± 3 °C/5 min before freeze-drying. Various structural properties were investigated to elucidate the varietal and processing-induced effect on molecular and secondary structure. Irrespective of processing, isolated proteins had a similar molecular size, with α-conglutin (412 kDa) and β-conglutin (210 kDa) being principal fractions for the albus and angustifolius variety, respectively. Smaller peptide fragments were observed for the pasteurized and spray dried samples, indicating some degree of processing-induced changes. Furthermore, secondary structure characterization by Fourier-transform-infrared and circular dichroism spectroscopy showed β-sheet and α-helical structure being the dominant structure, respectively. Thermal characterization showed two denaturation peaks corresponding to β-conglutin (T_d_ = 85–89 °C) and α-conglutin (T_d_ = 102–105 °C) fractions. However, the enthalpy values for α-conglutin denaturation were significantly higher for albus species, which corroborates well with higher amounts of heat stable α-conglutin present. Amino acid profile was similar for all samples with limiting sulphur amino acid. In summary, commercial processing conditions did not have a profound effect on the various structural properties of lupin protein isolates, and properties were mainly determined by varietal differences.

## 1. Introduction

The global demand for alternative or complementary proteins is increasing as traditional animal sources of protein are not only inefficient but may also have a detrimental effect on the environment [1]. Proteins from plant sources are preferred due to their low impact on the environment, the consumer concern of zoonosis, and consumer shifts (especially millennials and ‘Gen Z’) looking to either reduce meat consumption or switch to more plant-forward options [2,3,4]. Despite the largely favourable characteristics of soy and pea protein in the global market, there is not a ‘perfect’ food or crop, and soy and pea are not without their limitations. Allergenicity and the genetically modified status of certain soy varieties, along with some limitations in the functional properties of pea protein warrant the search for further options, such as lupin. However, a wider uptake of these novel sources of protein by food processors and consumers requires detailed structural and functional characterization.

Lupin is historically consumed throughout the Mediterranean and Andean regions and is a major legume crop grown in Australia, with a global annual production of over 1 million metric tonnes [5]. Lupin is densely nutritious given its high protein and fibre content, low to no starch, high unsaturated fatty acids, and high dietary polyphenols, which are associated with various health benefits [6,7,8]. In addition to being a rich source of protein (40–45%), lupins are sustainability champions because of their ability to fix higher amounts of nitrogen and grow in diverse climatic conditions of drought and saline, and they have cheap production cost compared to soy [8]. Of the four domesticated varieties, *Lupinus angustifolius* and *Lupinus albus* are grown widely in Australia, with these two varieties also being the most dominant worldwide, a testament to the low alkaloid content in these species [9,10]. Lupins are traditionally used as feed and forage; however, with the breeding success to obtain a low alkaloid lupin, more research and industry attention is currently focused on the extraction of proteins for human consumption.

Lupin protein is comprised of globulin and albumin in the ratio of 9:1, with four main globulin fractions identified as α-conglutin (legumin-like, 11S, 330–440 kDa), β-conglutin (vicilin-like, 7S, 143–260 kDa), γ-conglutin (7S, 200 kDa), and δ-conglutin (2S, 13 kDa) [11,12]. Of the four conglutins, α and β conglutin together make up to 90% of total globulins, while the γ and δ constitute the remaining fraction. Extraction and post-processing condition (e.g., drying) alter both the quantity and nature of the protein fraction. For example, alkaline and salt extraction followed by isoelectric and dilutive precipitation of proteins from *L. angustifolius* showed fundamentally different microstructural organization, with alkaline extraction resulting in the higher unfolding of protein while salt extraction preserved the ordered structure [13]. Similarly, water soluble δ-conglutin was lower in salt-extracted samples, while alkaline extract had a lower γ-conglutin [11]. Different protein fractions in the protein isolate give unique structural and functional attributes to lupin protein during the formulation of food and beverages and therefore warrant a detailed study of the structure.

Alkaline extraction followed by iso-electric precipitation remains the most widely adopted method for protein extraction, given its simplicity and higher yield [7,14]. Albe-Slabi et al. used alkaline (pH 7) and acidic conditions (pH 2.0) to extract proteins from *L. albus* with the acidic extract resulting in denatured proteins, with low thermostability and altered secondary structure [15]. Regarding other downstream processing methods, such as pasteurization, and different drying techniques, such as freeze drying or spray drying, although used in conjunction with the extraction and isolation process, no attention has been given to study the effect of these processing steps on the structural and functional properties of protein isolates. For example, pasteurization is extensively used by food industries to kill microorganism that are deemed unsafe for human consumption; however, to the best of our knowledge, no work has been done to study the effect of high temperature and short time pasteurization on lupin protein isolates. Furthermore, heat treatment has been known to cause protein denaturation and insolubilization [16]. The primary aim of this study was to study the effect of drying conditions on the structural properties of lupin protein isolates from two major Australian lupin species. Given that most industries use pasteurization to kill pathogenic microorganism, a secondary aim of this study was to assess the effect of pasteurization on protein structure in contrast to direct freeze drying.

## 2. Materials and Methods

### 2.1. Materials and Chemicals

Commercial lupin flours of *L. angustifolius* cv Jurien and *L. albus* cv Murringo were purchased from Australian-based suppliers ‘Lupin Co’ and ‘Lupin for Life’, respectively. For simplicity, Jurien (J) and Murringo (M) will be used to denote angustifolius and albus lupin cultivars, respectively, throughout the manuscript. The whole flours were used as received for protein extraction. For defatted flours, 100 g of flour was mixed with 600 mL of hexane with continuous mixing for 6 h. The hexane layer was removed, and the flour was again mixed with 300 mL fresh hexane and extracted for 3 h before drying at 40 °C overnight.

For protein extraction and all other analysis, Milli-Q^TM^ water (Millipore SAS, Molsheim, France) was used. Analytical grade chemicals and reagents including hydrochloric acid (HCl), sodium hydroxide (NaOH), sodium phosphate monobasic dihydrate (NaH_2_PO_4_·2H_2_O), sodium phosphate dibasic dihydrate (Na_2_HPO_4_·2H_2_O), and sodium chloride (NaCl) were purchased from Sigma-Aldrich (Melbourne, Australia). Details of the reagents for sodium dodecyl sulphate-polyacrylamide gel electrophoresis (SDS-PAGE) and size exclusion chromatography (SEC) are mentioned in the respective sections.

### 2.2. Protein Extraction

Alkaline extraction (pH 8.0) followed by isoelectric precipitation (pH 5.0) was used for preparation of protein isolates, as shown in Figure 1. Either full fat or lab defatted flour was taken and suspended in Milli Q water at a flour to water ratio of 1:10. An alkaline pH of 8.0 was maintained by adding small quantities of 1 M NaOH until the pH was achieved. Protein was extracted for 2 h with continuous stirring at room temperature (~23 °C). The protein rich fraction, i.e., the supernatant, was then collected after centrifugation and precipitated at pH 5 overnight at 4 °C. The extraction and precipitation pH were optimized based on several preliminary experiments. Extraction pH of 8, 9, and 10 were tested with no significant difference in protein yield at these conditions. Therefore, pH 8.0 was selected, which would require less alkali to be added. Similarly, a precipitation pH of 4, 5, and 6, individually as well as sequentially, in different combinations was tested. This resulted in pH 5.0 having the highest yield, at least in number of steps. The precipitate was further purified by centrifugation at 15,000× *g* for 10 min at 4 °C, followed by washing with MilliQ water. The resulting protein isolates were then resuspended in MilliQ water to obtain a solid concentration of approximately 20–30%, and pH was maintained at 7.2 ± 0.1 using 0.1 M NaOH. The slurry was then either freeze or spray dried directly or was pasteurized at 75 ± 3 °C for 5 min and freeze dried. The pasteurization temperature and time were modified based on the milk pasteurization condition (72 °C/15 s). However, given the higher solid content (20–30%) of the protein isolate slurry, the time–temperature combination was slightly increased. This was sufficient to kill all pathogenic microorganisms (no Salmonella) and reduce non-pathogenic microbial load to an acceptable limit (Total plate count <1000 cfu/g, yeast and mould <50) and to not have a detrimental effect on the product properties. Spray drying was performed using a Buchi Mini Spray Dryer B-290 (Buchi AG, Flawil, Switzerland) with inlet and outlet temperatures of 180 °C and 85 °C, respectively. The feed slurry concentration was 20–30% solids.

### 2.3. Proximate Analysis

The moisture content of the samples was determined using a method described by AOAC (2005) [17]. The sample (1 g) was placed in crucibles, following which its weight was recorded. The crucible with the sample was then heated at 105 °C utilla constant weight. The dried crucibles were then placed in desiccators and reweighted on reaching room temperature.

The fat content of the sample was determined by using an acidified alcohol hydrolysis method followed by solvent extraction in a hexane and ether mixture (AOCS 922.06) [18]. The sample (2 g) was first hydrolysed using ethanol and 8 M HCl with heating at 70 °C for 30 min. The fat was then extracted using a mixture of hexane and ether. The solvents were then evaporated and the remaining content weighed for total fat.

The ash content was determined in a thermo-gravimetrical system by combustion at 950 °C until constant weight according to AACC 08-01 [19].

The protein content was determined using the Kjeldahl method (AACC 46-12) [19]. The protein content was calculated based on the nitrogen-to-protein conversion factor of 5.7 used for seed storage protein (and N × 6.25 to facilitate comparison with the literature) [13,20]. A protein conversion factor of 6.25 is generally used for animal source protein, assuming a nitrogen content of 16%. However, for grains, a protein conversion factor of 5.7 is used to avoid overestimation of protein content due to the presence of non-protein nitrogen content [21].

### 2.4. Sodium Dodecyl Sulphate-Poly Acrylamide Gel Electrophoresis (SDS-PAGE)

SDS-PAGE was performed using Bio-rad criterion Cell Vertical Midi-format electrophoresis cell (Bio Rad Laboratories Inc., Hercules, CA, USA), with all chemicals purchased from Bio-Rad. Firstly, samples (1 mg/mL) were prepared in phosphate buffer saline solution (1 mM, pH 7.4) and were left to mix for 2 h at room temperature in a tube roller. The supernatant was then collected after centrifugation at 10,000× *g* for 15 min at 20 °C. A 30 µL sample was then mixed with 10 µL of 4× Laemmli sample buffer. β-mercaptoethanol was mixed with sample buffer as a reducing agent for reducing samples. A 20 µL sample was loaded into Bio-rad 4–20% criterion TGX precast gels, and Tris/Glycine/SDS was used as the running buffer. Gels were run for ~40 min at 200 V power. The gels were then stained using Coomassie brilliant blue staining solution, which was then followed by de-staining solution, both supplied by Bio-rad. A picture of clear gels with protein bands was then taken using a standard camera gel band photography.

### 2.5. Size Exclusion Chromatography

Size exclusion chromatography of protein isolates was performed using an AKTA Pure 25 protein purification system (AKTA Pure, GE Healthcare, Diegeum, Belgium) fitted with a superdex 200 increase 10/300 GL column (Cytiva, Little Chalfont, UK). Samples (40 mg/mL) were dissolved in 10 mM phosphate buffer containing 150 mM NaCl (pH 7.2) and shaken in a tube roller for 2 h, which was then centrifuged at 10,000× *g* for 15 min. The supernatant was then filtered through a 0.2 µm polyether sulphone filter. A 0.5 mL aliquot of the sample was injected into the column, and the phosphate buffer (10 mM, 150 mM NaCl) was passed through the column at a constant flow rate of 0.5 mL/min. UV absorbances were collected for 280 nm. Molecular weight standard 15–670 kDa (Sigma Aldrich, Melbourne, Australia) was used for calibration. A calibration curve was constructed by plotting elution time against the log molecular weight of the protein standard. The linear equation thus obtained was to calculate the molecular weight for known elution times.

### 2.6. Amino-Acid Quantification by Ultra-High-Performance Liquid Chromatography (UHPLC)

Samples (20–30 g) were digested at 110 °C for 16 h in either acid (2 mL of 6 M HCl with 0.1% phenol) or alkali (3 mL of 4 M LiOH containing 95 mM ascorbic acid). Air in glass vials was replaced by flushing nitrogen gas, and the vials were immediately sealed. After digestion, the samples were neutralized with freshly prepared alkali (2 mL of 6 M NaOH) or acid (6 M HCl), depending upon initial digestion. The pH was maintained around 6–8 with pH strips. The total volume was then made up to 10 mL using MilliQ water. The digested samples were then centrifuged at 13,000× *g* for 5 min, and the supernatant was filtered via a 0.2 µm syringe filter. The filtrate aliquots were subjected to further derivatization following the protocol described by Valgepea et al. [22]. A Vanquish Core UHPLC (Thermofisher Scientific) equipped with a Vanquish Fluorescence detector and high performance autosampler with online derivatisation was used for chromatographic identification and quantification of amino acids. A calibration curve was created using serially diluted amino acid standard mixture (Sigma AAS 18–10 mL) and amino acid supplement (Agilent, 5062-2478) kits. The upper and lower limits of quantification were 500 and 1.95 μM, respectively (except for proline: 3.9–1000 μM). Amino acids were then expressed as g/100 g protein.

### 2.7. Secondary Structure Characterization by Fourier Transform Infrared and Circular Dichroism Spectroscopy

Fourier transform infrared (FTIR) spectroscopy was used to assess the secondary structure of proteins. FTIR spectra for protein isolates were obtained using a Perkin Elmer Spectrum 2 FTIR fitted with a diamond crystal universal attenuated total reflectance (UATR) accessory (Perkin Elmer, Shelton, CT, USA). The dried sample was placed onto the diamond and pressed using a pressure knob and shoe press. Spectra were collected in the mid-range infrared region from 4000 to 400 cm^−1^ at a resolution of 4 cm^−1^ and 32 scans. Spectra were obtained for duplicate samples and averaged.

Secondary structure was determined using a Gaussian curve fitting of the after deconvolution of the Amide I region (1600–1700 cm^−1^) using the Microsoft Excel solver function. Each peak was assigned to its corresponding structure according to previous studies [23,24]. The integral area of each peak was divided by the sum of all determined peaks to calculate the relative area of specific secondary structure.

Circular Dichroism (CD) spectra in the far UV range (190–260 nm) were collected using a spectropolarimeter (Jasco J-815, Jasco International Co., Ltd., Tokyo, Japan) and a rectangular cell (21/10/CD/Q1, Starna Pty. Ltd., Castle Hill, NSW, Australia), with an optical path length of 1mm. A protein dispersion of 1 mg/mL was first dissolved in 10 mM phosphate buffer solution (no salt) and stirred for 2 h. The dispersion was then centrifuged for 30 min at 16,000× *g*, and the supernatant was filtered through a 0.45 µm syringe filter. The protein solution was diluted (~0.2 mg/mL) before obtaining spectra to keep the voltage below 700. Then, 400 µL of the sample was loaded onto the measurement cell, and spectra were collected at 20 °C at a scan rate of 100 nm/min, bandwidth 1 nm, and a spectral resolution of 1 nm. Corrected and averaged molar ellipticity from three measurements were then analysed for secondary structure using the CONTIN-LL algorithm, available at the DichroWeb website (http://dichroweb.cryst.bbk.ac.uk/html/process.shtml, accessed on 18 November 2021).

### 2.8. Thermal Characteristics

The thermal characteristics of lupin protein isolates were analysed using differential scanning calorimetry (DSC) (TA DSC 2500 TA instruments, New Castle, DE, USA). Samples were first dispersed in deionized water (20% *w*/*w*) by stirring at room temperature for 1 h. The protein solution was then heated in hermetically-sealed DSC aluminium pans (Tzero, TA Instruments, New Castle, DE, USA), using an empty pan as reference. Thermograms were obtained by linear heating from 20 °C to 120 °C at a heating rate of 10 °C min^−1^. Peak denaturation temperature (T_d_) and the enthalpy of denaturation ΔH (J/g) was calculated automatically by the TA Universal Analysis software (version 5).

### 2.9. Data Analysis and Statistics

Experiments were undertaken with at least duplicate samples, and results are presented as average ± standard deviation. Statistical analysis was performed using IBM SPSS Statistics software (Version 26, 2019). Where possible, data were subjected to one way-ANOVA analysis, and differences in means were compared at *p* < 0.05 significance level using Tukey B post hoc analysis test. Figures were drawn using SigmaPlot 14.5 software. Principal component analysis was performed using XLSTAT-student 2022.

## 3. Results and Discussion

### 3.1. Protein Extraction and Proximate Analysis

Alkali solubilization followed by isoelectric precipitation was used to isolate proteins from two prominent Australian lupin varieties. A general process flow diagram for protein extraction from lupin is presented in Figure 1. For ease of following, *L. angustifolius* cv Jurien will be denoted by Jurien (J) and *L. albus* cv Murringo will be denoted by Murringo (M) throughout this article. The main difference in processing was the drying step, with one group of samples being freeze dried (−80 °C, sublimation) while another group was spray dried (T_in_ = 180 °C, T_out_: 85 °C). One of the groups was also pasteurized at 75 ± 3 °C for 5 min before freeze drying. The pasteurization time and temperature combination was selected based on preliminary experiments, which were sufficient to kill all pathogenic microorganisms and reduce other microorganisms to a low acceptable limit, i.e., total plate count less than 1000 cfu/g and yeast and mould count less than 50. Pasteurization or mild heat treatment is often used by industrial protein processors to kill pathogenic microorganism. Extraction of proteins at high temperature (~60 °C) is also common as it is found to increase extraction yield in addition to preventing microbial growth. However, extraction of soy protein isolate at 80 °C for 30 min yielded significantly less proteins and also reduced emulsion capacity and solubility [25]. Prolonged exposure at elevated temperature is costly and will also have detrimental effect on protein functionality. Freeze drying and spray drying are the most common methods of preserving protein for longer storage, with moisture content being lower than 8%. Freeze drying is often used in research and laboratory-based studies, while spray drying is used industrially.

The proximate composition of protein Isolates obtained at different processing regimes is presented in Table 1. Protein content varied from 80–95% (N × 6.25), with the defatted samples resulting in significantly higher protein content compared to the original flour. In terms of processing conditions, freeze drying followed by pasteurization and spray drying resulted in similar protein concentration (Table 1), with one exception of full fat angustifolius variety showing significantly higher protein content compared to its freeze-dried counterpart. In addition, among different treatments across same raw material, direct freeze-dried samples had the highest protein content, although not statistically different.

Protein content (N × 5.7) in all isolate was comparable to previously reported studies, although the full-fat isolates had a slightly lower protein concentration [13,26], with the fat content being higher than previously reported values for defatted samples. Although N × 6.25 showed higher protein concentration, this might be due to the overestimation of nitrogen, considering the non-protein nitrogen. Higher fat is deemed undesirable as it may affect longer term storage of proteins via oxidation. On a brighter note, however, having a good amount of fat is desirable in products such as plant-based milk or dairy mimic formulation as it gives mouthfeel and other organoleptic properties.

### 3.2. Protein Molecular Size Determination Using SDS-PAGE

Separation of the complex protein mixture was performed using a sodium dodecyl sulphate polyacrylamide gel electrophoresis (SDS-PAGE), which uses sodium dodecyl sulphate to linearize and eliminate charge, thus resulting in separation of proteins solely based on size. The separated proteins were retained on the gel network and compared to the known size of protein markers to determine the size of the protein fraction. The use of a reducing agent such as β-mercaptoethanol breaks the di-sulphide bond present in some fractions of lupin conglutin proteins. The SDS-PAGE profile of the lupin protein isolates produced by different conditions is shown in Figure 2a, for non-reducing, and Figure 2b, for reducing conditions.

The molecular size of proteins in lupin isolates had a broad range from 75–10 kDa, with both varieties of lupin having some similar as well as distinctive protein bands. On the non-reducing SDS-PAGE, the albus variety had two distinct bands around 45 kDa and 38 kDa, whereas the angustifolius variety had distinct bands at 75 kDa, 55 kDa, and 20 kDa; faint bands were also observed for the albus variety at 38 kDa. Clear protein bands were observed for both varieties in the region of 65 kDa, 32 kDa, 18 kDa, and 15 kDa on the non-reducing and 65 kDa, 48 kDa, 37 kDa, 32 kDa, 18 kDa, 15 kDa, and 10 kDa on the reducing SDS-PAGE. Lupin proteins are known to have four fractions based on their electrophoretic mobility, with α- and β-conglutin being the most dominant globulins, comprising about 80% of proteins, followed by γ-conglutin (15–20%) and albumin-like δ-conglutin comprising about 5% [12]. Among the four fractions, β-conglutin exists as trimers without a di-sulphide bond, with monomeric units ranging in size from 68 to 75 kDa. Alpha conglutin, on the other hand, exhibits as a hexamer with six disulphide bonds, which are broken by reducing agents to result in monomeric units of 58, 74, and 67 kDa [13]. Monomeric units of β- and α-conglutin are more likely to be seen in Figure 2. On both non-reducing and reducing SDS-PAGE, β-conglutin is observed at approximately 65 kDa and 32 kDa, while α-conglutin is observed at 75 and 55 kDa on non-reducing electrophoresis and ~46 kDa on reducing SDS-PAGE. Several smaller fragments were also observed under reducing conditions at 12 and 10 kDa, which can be associated with δ-conglutin fractions, and 18 kDa, which is a broken down fraction of γ-conglutin. A similar protein profile was reported for isolates from *L. angustifolius* by Muranyi et al. [11] and for *L. albus* by Duranti et al. [12,27]. Processing does not seem to have a major effect on protein profile, with identical profiles being observed for all types of processing, contrary to previous reports stating the complete denaturation of proteins in commercially processed samples [28]. Smaller fractions of ~10 kDa were observed, even at non-reducing conditions, especially for the pasteurized sample, with the degradation more likely caused by heat. Under the non-reducing condition, some bands were more clear than others for specific variety, such as the ~45 kDa band for the albus variety being more pronounced than that on the angustifolius and bands at 75, 55, and 18 kDa more visible for angustifolius than on albus, which can be due to varietal differences in protein. However, the same cannot be said on the reducing conditions, where bands were identical for both verities.

### 3.3. Protein Size Determination Using Size Exclusion Chromatography (SEC)

Size exclusion chromatography (SEC) was performed to obtain the size of native lupin proteins in protein isolates obtained by different processing techniques. Unlike SDS-PAGE, SEC gives the molecular size of soluble proteins in their native state. Size exclusion chromatograms from *L. angustifolius* cv Jurien and *L. albus* cv Murringo are shown in Figure 3a,b, respectively.

Similar protein sizes were obtained for both the angustifolius and albus variety. The albus variety showed some protein aggregates above a 650 kDa size, while the angustifolius did not have those aggregates. For angustifolius varieties, the legumin-like α-conglutin of 412 kDa was observed at about 9 mL elution volume, while vicilin-like β-conglutin of 216 kDa were observed at 11 mL elution volume; a small fraction of the low molecular weight protein of 38 kDa was observed around 15 mL elution volume, which can be attributed to δ-conglutin. Similarly, for the albus variety, large protein aggregates of >670 kDa were observed at around 8 mL elution volume, while α-conglutin of 394 kDa, β-conglutin of 210 kDa, and δ-conglutin of 30–38 kDa were observed at 9, 11.5, and 15.5 elution volumes, respectively. In both varieties, no γ-conglutin peaks can be observed, which can be linked to the acidic precipitation method (pH 5.0) used in this study, contrary to an alkaline (~7.9) isoelectric point of γ-conglutin protein fraction [11,12]. Alpha and β-conglutins in their native state exist in hexameric and trimeric form, and their molecular weights are noted in the range of 330–430 and 143–260 kDa, respectively for albus lupins [12,29], which is concurrent to the current study. A similar protein profile was reported for *L. mutabilis* with β-conglutin of 224 kDa, α-conglutin of 226 kDa, and γ-conglutin of 63 kDa [30]. Although not quantified, for both varieties, peak UV absorbance was lower for pasteurized samples and was the highest for direct freeze-dried samples, indicating lower quantities of protein fractions and thus some degree of change due to processing. Furthermore, absorbance attributed to smaller peptide fractions, <10 kDa, were higher for pasteurized and spray-dried samples compared to those of direct freeze-dried samples, indicating a higher degradation of proteins due to heat treatment. In summary, both varieties had similar protein profiles with different processing techniques; in particular, the application of thermal treatment caused a change in protein concentration and resulted in smaller peptide fractions. One notable difference was that legumin-like α-conglutin (412 kDa) was a dominant fraction in th albus variety, while the vicilin-like β-conglutin (210 kDa) was the dominant protein in the angustifolius variety. This is also validated by SDS-PAGE results, wherein bands for α-conglutin (~75 kDa, 55 kDa) were more pronounced for angustifolius, while the β-conglutin bands (45 kDa) were more noticeable for the albus variety. The legumin-to-vicilin ratio is an important feature of plant proteins and has been linked to various structural and functional attribute of the protein [31]. For example, legumin-like proteins are known to form strong and stable gels, while vicilin-like fractions form better and stable emulsions [32].

### 3.4. Amino Acid Quantification of Protein

Amino acid composition for lupin protein isolates obtained by different processing techniques is shown in Table 2. Both varieties of lupin isolate had almost similar amino acid profiles, with the variation coming from processing techniques. Glutamic acid was the most abundant amino acid in all samples, with values ranging from 23 to 27%, followed by similar quantities of arginine and aspartic acid (10–13%). As previously reported for lupins [33,34] and other legumes [35], sulphur-containing amino acids were in low quantity (~1% cysteine and ~0.5% methionine), and the values reported were in line with this work. The total essential amino acid was approximately 27–33%, similar to those values reported by Sujak et al. [33]. In accordance with FAO/WHO guidelines, the total amino acid was slightly lower than the reference value of 36%, while total aromatic was abundantly rich (>6%). Similarly, lysine, leucine and isoleucine values were also higher than the reference values [36]. Total amino acids within the same functional group also showed a similar trend, with the freeze-dried Jurien cultivar showing significantly low total aromatic amino acids.

The amino acid side chain of a protein gives specific functional and structural properties to the protein. For example, a higher amount of non-polar amino acid gives a higher hydrophobicity, and vice versa, which is reflected on its functional properties, such as gelling, rheology, and solubility [37]. Total basic amino acids, which are expected to be high in alkaline-extracted protein, are also similar, with slightly higher values observed in angustifolius variety. Both polar and non-polar amino acids were similar for all samples in this study, thus limiting our ability to study the effect. On the other hand, no direct correlation can be drawn between processing techniques and amino acid profile. The underlying explanation for this could be that all extractions and isolations were performed following identical protocols, while only the drying process was different.

### 3.5. Secondary Structure Determination Using FTIR and CD

The FTIR spectroscopy of proteins obtained in the amide I region (1700–1600 cm^−1^) can be related to the protein secondary structure, such as the α-helix, β-sheet, β-turns, and unordered structures. The secondary structure of lupin proteins obtained by different processing techniques is presented in Figure 4. The FTIR spectra of the amide I region and deconvoluted peaks corresponding to the secondary structure is shown in Appendix A. Four prominent secondary structures were observed on both FTIR and CD spectroscopy results.

The FTIR spectra analysis revealed gaussian bands with mid-point in the region of ~1610, ~1685, ~1637, ~1656, and ~1663 cm^−1^ corresponding to the A1, A2, β-sheet, α-helix, and β-turn structures. The A1 and A2 confirmation can be attributed to amino acid side chains and protein aggregates [38]. All protein samples had the highest amount of β-sheet structure (30–55%), followed by aggregates (A1 and A2 combined), 13–35%, while α-helices were in low quantities, 0–35%. Beta sheet structures are known to have peaks centred at 1625 cm^−1^ (1612–4641), 1633 cm^−1^ (1626–1640), and 1682 cm^−1^ (1670–1694) [38,39]. Two peaks at ~1625 cm^−1^ and ~1638 cm^−1^ were assigned to the β-sheet structure for lupin protein isolates. The β-sheet structure is reported to be the most dominant secondary structure of the legume protein, which corroborate well with this study. Beta-sheet and β-turns accounted for approximately two-thirds of the secondary structure observed in lupin protein isolates. Processing also affected the secondary structure composition, with direct freeze-dried samples showing higher retention of α-helical structure.

CD spectra provides highly sensitive information about the secondary structure of soluble proteins. The CD-derived secondary structure of soluble lupin protein isolates is presented in Figure 4b, with representative CD spectra shown in Appendix A. In contrast to the FTIR results, the secondary structure of soluble proteins was dominated by a random or unordered structure and a lower β-sheet value, apart from the direct freeze dried and spray dried defatted samples of both lupin varieties, which showed an approximately ~40% (angustifolius) and 30% (albus) β-sheet structure. A more heat-labile α-helix was the second most dominant structure, with values ranging from as low as 10% (JD-S) to as high at 50% (M-S). Mildly processed samples such as direct freeze- and spray-dried samples had a higher proportion of β-sheet structures, especially for defatted samples. For heat-treated samples, pasteurized samples had a higher proportion of heat-labile α-helix and random order structure, indicating some degree of secondary structure change. Albe-Slabi et al. [15] showed that the CD-derived secondary structure for lupin proteins isolated at different pH was dominated by a α-helix structure when isolated at a neutral pH of 7.0, while at pH 2.0 the secondary structure was dominated by a β-sheet structure. Furthermore, in other research, the effect of a pasteurization temperature of 65 °C for 30 min or 85 °C for 2 min did not have any effect on the secondary structure of lentil proteins [40].

### 3.6. Thermal Characterization Using Differential Scanning Calorimetry

The thermal properties of proteins were determined using differential scanning calorimetry (DSC), and the obtained thermograms for *L. angustifolius* and *L. albus* lupins are presented in Figure 5a,b, respectively. DSC provides valuable information regarding the denaturation temperature (T_d_) and enthalpy (∆H), which can be related to the thermal stability of proteins and the degree of denaturation, as well as the extent of native structures. All lupin isolates showed two T_d_, the first at 88–89 °C and the second at 102–105 °C, with a distinct second peak for the angustifolius variety compared to the albus. The two peaks observed in lupins are associated with two different fractions of lupin globulin proteins, i.e., the legumin-like α- and the vicilin-like β-conglutins [16,41,42], with the first peak representing the β-conglutin, which is also the dominant fraction in both lupin variety. A single peak for β-conglutin was reported for albus lupins [41], which corroborates well with the small second peak height observed for albus lupins. Protein denaturation causes change in its 3D configuration as a result of intra- and inter-molecular cleavage. The vicilin-like β-conglutin in lupin is mainly bound by weak non-covalent interaction and is thus easily broken compared to the di-sulphide linkages in α-conglutin, which require more heat energy [16]. ∆H values varied from 2.29 to 6.80 J/g for angustifolius in the first peak to 1.06 to 4.74 J/g for second peak, while the same for albus variety ranged from 4.63 to 7.08 for first peak to 0.29 to 0.6 J/g for second peak. These values are higher than the previously reported values of <3 J/g [16,41], indicating that the processing conditions did not have a harsh effect on protein quality and functionality. While the peak denaturation temperature was similar for all samples, enthalpy values were particularly lower for the pasteurized sample, indicating that the proteins are denatured to a higher extent. However, the pasteurization temperature used was lower than the actual denaturation temperature of lupin protein and therefore the changes may not solely be due to the heat treatment used. Although the spray drying temperature was higher than the denaturation temperature, the enthalpy was higher for sthe pray-dried samples. This can be justified by the lower residence time during spray-drying conditions.

### 3.7. Relationship between Different Structural Properties

Principal component analysis (PCA) provides valuable information on the relationship between various parameters with independent variables. PCA reduces the number of variables into a small, consolidated set, therefore helping to find a clearer relationship between variables. A principal component analysis loading plot and biplot showing the relationships between the different structural properties of protein isolate are shown in Figure 6a,b, respectively. The loading plot shows a strong positive correlation between component 1 and the beta structure from CD, total non-polar amino acid, unordered structure from FTIR, denaturation temperature of β-conglutin, and enthalpy of β-conglutin denaturation. Both the loading and biplot showed varietal differences with the Murringo variety, associated with component 1, while Jurien were more closely related to component 2. On the biplot, the smaller angle between variables indicates a higher association. The PCA and biplot showed a strong relationship between α-helix from CD and FTIR; however, the β-sheet and unordered structures were not correlated. This could be explained by the soluble fraction used for CD spectra analysis, while the whole insoluble and soluble fraction are used for FTIR spectra. In addition, protein was closely related to the enthalpy and denaturation temperature, as well as with β-sheet structure and total non-polar amino acid. In Figure 6b, the clustering of different protein isolate samples can be seen. It is evident from this figure that the clustering was more related to varietal differences than the processing techniques used. The full fat Jurien cultivar grouped together as one cluster, while the Murringo cultivar clustered together as another group. Spray-dried and direct freeze-dried samples, especially the defatted ones (MD-S and MD-F, JD-S and JD-F on Figure 6b), were seen to form another cluster, while the full fat freeze dried samples were close to their pasteurized and spray-dried cluster.

## 4. Conclusions

The structural properties of lupin protein isolates obtained by varying processing techniques showed minimal effects of processing on the structural behaviour. Small differences can be attributable to varietal differences, as opposed to the processing steps alone. Firstly, a similar protein profile was observed on SDS-PAGE and SEC for both varieties, showing the presence of α-, β-, and δ-conglutins, with, however, few distinct non-overlapping bands for both varieties. During the SEC profile analysis, a more prominent peak was observed for β-conglutin for albus lupin, which later correlated well with a higher peak enthalpy value for albus lupin. A smaller fragment of peptides was more evident on the SEC for the pasteurized samples compared with the direct freeze-dried samples. Secondary structure quantification by FTIR showed the highest amount of β-sheet structure, while CD showed the highest quantities of random or unordered structure. Alpha helical structure correlated well between the FTIR and CD obtained values. Furthermore, thermal characterization by DSC showed two distinct peak denaturation temperatures, corresponding to the α- and β-conglutin globulin fraction present in the lupin isolates. Higher enthalpy values for β-conglutin denaturation for albus lupin indicate a higher stability of these protein and are not denatured highly, even at the pasteurization temperature used. Finally, principal component analysis revealed a compositional clustering of protein isolates rather than the processing techniques used. It can also be concluded that the mild processing conditions used in this study did not have a significant effect on the structural properties of lupin isolates obtained from two different Australian varieties. The processing-induced changes on the structure of a protein will likely be reflected in its functional characteristics, which needs further investigation. Nevertheless, the findings of this study will help food processors to better design food processing systems that involve lupin proteins. This study will also provide the scientific community with knowledge regarding lupin protein structures and the way they are affected by commonly-used processing techniques.

## Figures and Tables

**Figure 1 foods-12-00908-f001:**
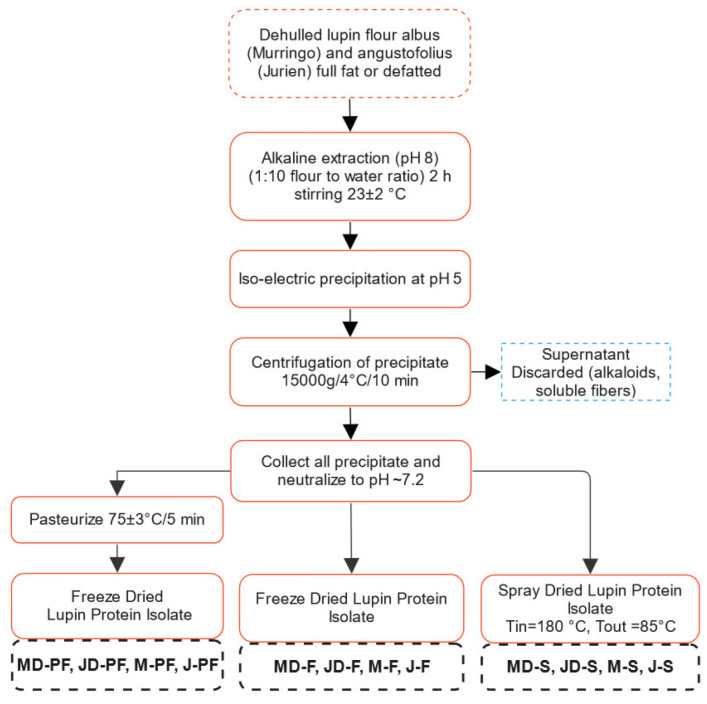
Flow chart showing different processing techniques used for obtaining lupin protein isolate. MD-F = Defatted Murringo direct freeze dried, JD-F = Defatted Jurien direct freeze dried, M-F = Full fat Murringo direct freeze dried, J-F = Full fat Jurien direct freeze dried, MD-PF = Defatted Murringo pasteurized and freeze dried, JD-PF = Defatted Jurien pasteurized and freeze dried, M-PF = Full fat Murringo pasteurized and freeze dried, J-PF = Full fat Jurien pasteurized and freeze dried, MD-S = Defatted Murringo spray dried, JD-S = Defatted Jurien spray dried, M-S = Full fat Murringo spray dried, J-S = Full fat Jurien spray dried.

**Figure 2 foods-12-00908-f002:**
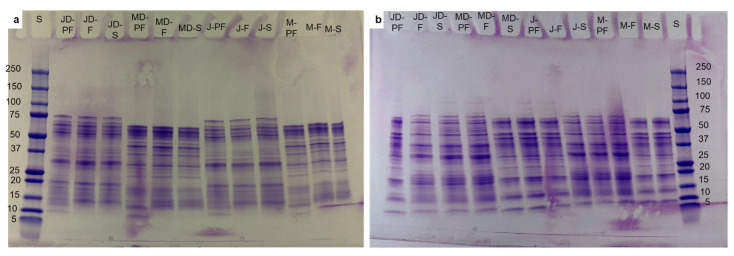
SDS-PAGE profile of lupin protein isolates obtained by different processing techniques: (**a**) without reducing agent, (**b**) with reducing agent (β-mercaptoethanol). S = Marker, MD-F = Defatted Murringo direct freeze dried, JD-F = Defatted Jurien direct freeze dried, M-F = Full fat Murringo direct freeze dried, J-F = Full fat Jurien direct freeze dried, MD-PF = Defatted Murringo pasteurized and freeze dried, JD-PF = Defatted Jurien pasteurized and freeze dried, M-PF = Full fat Murringo pasteurized and freeze dried, J-PF = Full fat Jurien pasteurized and freeze dried, MD-S = Defatted Murringo spray dried, JD-S = Defatted Jurien spray dried, M-S = Full fat Murringo spray dried, J-S = Full fat Jurien spray dried.

**Figure 3 foods-12-00908-f003:**
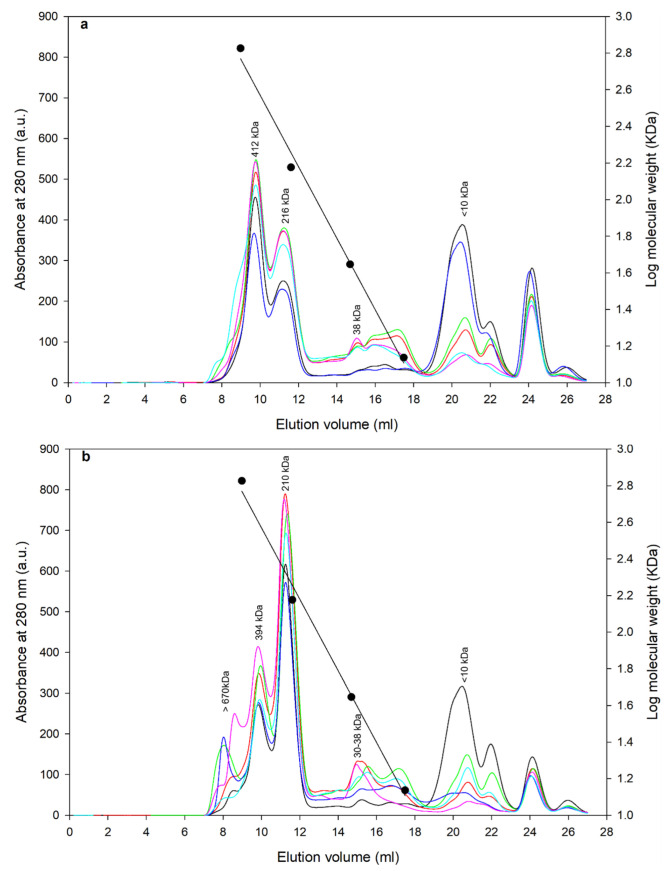
Protein peaks obtained by size exclusion chromatography (SEC) of lupin protein isolates prepared using different processing conditions; symbols are 

 defatted pasteurized, 

 defatted freeze dried, 

 defatted spray dried, 

 full fat pasteurized, 

 full fat freeze dried, 
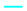
 full fat spray dried: (**a**) for *L. angustifolius* cv Jurien and (**b**) for *L. albus* cv Murringo.

**Figure 4 foods-12-00908-f004:**
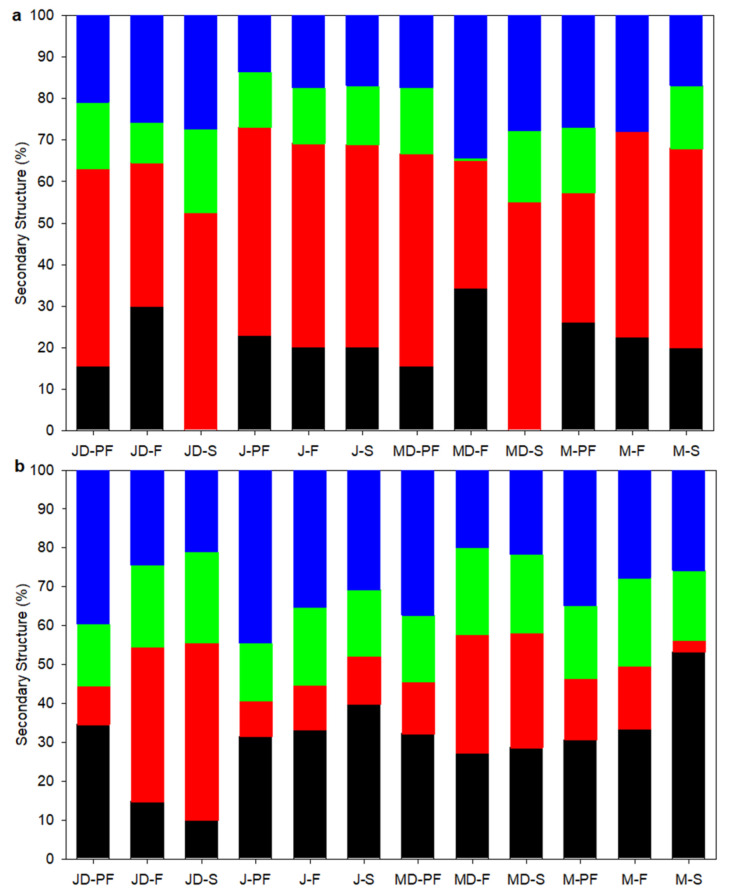
Secondary structure composition of lupin protein isolates obtained by different processing techniques: (**a**) using FTIR and (**b**) using CD. Symbols represent: 
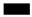
 α-helix, 
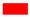
 β-sheet, 
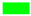
 β-turns, 
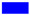
 Unordered/Aggregates (sum of A1 (~1610 cm^−1^) and A2 (~1685 cm^−1^) for FTIR). MD-F = Defatted Murringo direct freeze dried, JD-F = Defatted Jurien direct freeze dried, M-F = Full fat Murringo direct freeze dried, J-F = Full fat Jurien direct freeze dried, MD-PF = Defatted Murringo pasteurized and freeze dried, JD-PF = Defatted Jurien pasteurized and freeze dried, M-PF = Full fat Murringo pasteurized and freeze dried, J-PF = Full fat Jurien pasteurized and freeze dried, MD-S = Defatted Murringo spray dried, JD-S = Defatted Jurien spray dried, M-S = Full fat Murringo spray dried, J-S = Full fat Jurien spray dried.

**Figure 5 foods-12-00908-f005:**
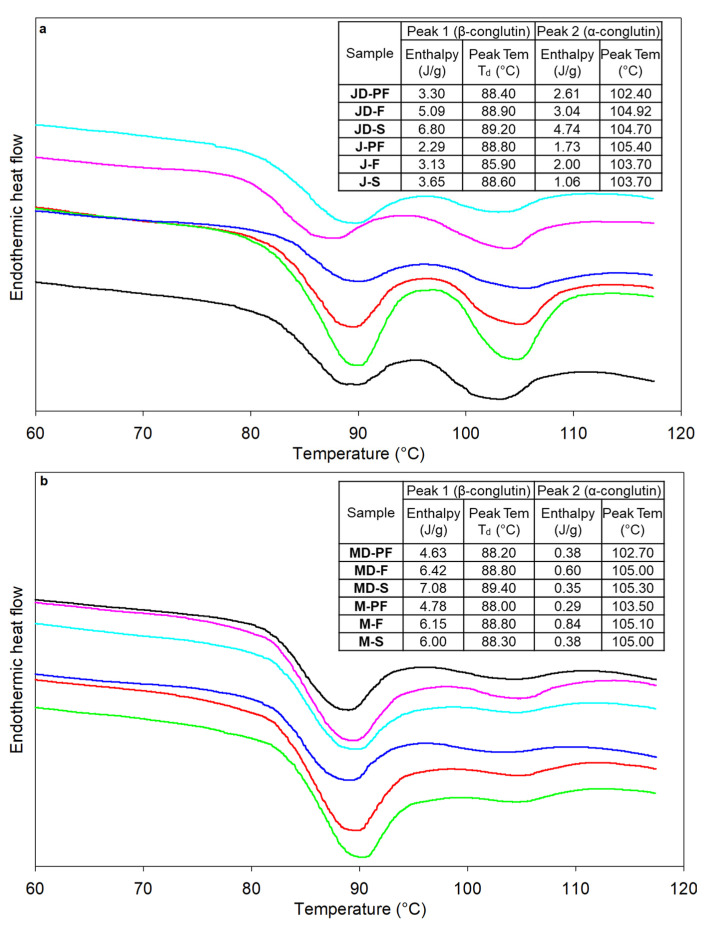
DSC thermograms for lupin protein isolates processed using different techniques (**a**) *L. angustifolius* cv Jurien and (**b**) *L. albus* cv Murringo. In the set are peak denaturation temperature and enthalpy; peak 1 corresponds to β-conglutin denaturation, while peak 2 corresponds to α-conglutin. symbols are 

 defatted pasteurized, 

 defatted freeze dried, 

 defatted spray dried, 

 full fat pasteurized, 

 full fat freeze dried, 

 full fat spray dried. MD-F = Defatted Murringo direct freeze dried, JD-F = Defatted Jurien direct freeze dried, M-F = Full fat Murringo direct freeze dried, J-F = Full fat Jurien direct freeze dried, MD-PF = Defatted Murringo pasteurized and freeze dried, JD-PF = Defatted Jurien pasteurized and freeze dried, M-PF = Full fat Murringo pasteurized and freeze dried, J-PF = Full fat Jurien pasteurized and freeze dried, MD-S = Defatted Murringo spray dried, JD-S = Defatted Jurien spray dried, M-S = Full fat Murringo spray dried, J-S = Full fat Jurien spray dried.

**Figure 6 foods-12-00908-f006:**
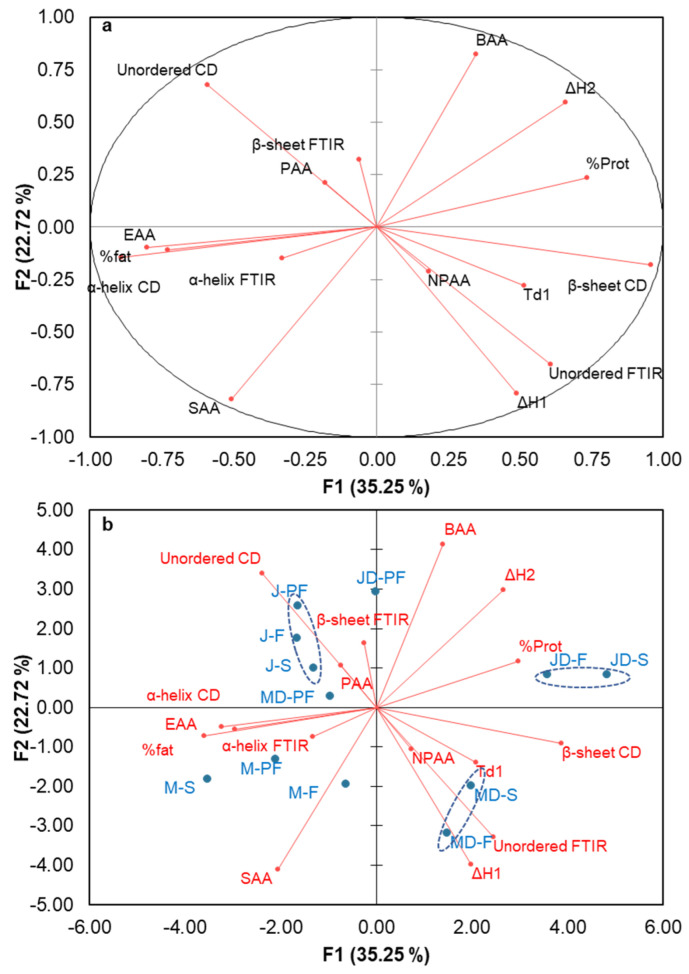
Principal component analysis: (**a**) PCA Loading plot, (**b**) biplot showing relationship between the various structural properties of lupin protein isolates obtained by different processing techniques. SAA = sulphur containing amino acids, EAA = essential amino acid, PAA = polar amino acids, NPAA = non-polar amino acids, BAA = basic amino acids, α, β-FTIR, α, β-CD = α helix and β-sheet from FTIR and CD, % Prot = protein concentration, Td1 = peak denaturation temperature for β-conglutin, ΔH1 and ΔH2 = enthalpy change for β and α-conglutin respectively. MD-F = Defatted Murringo direct freeze dried, JD-F = Defatted Jurien direct freeze dried, M-F = Full fat Murringo direct freeze dried, J-F = Full fat Jurien direct freeze dried, MD-PF = Defatted Murringo pasteurized and freeze dried, JD-PF = Defatted Jurien pasteurized and freeze dried, M-PF = Full fat Murringo pasteurized and freeze dried, J-PF = Full fat Jurien pasteurized and freeze dried, MD-S = Defatted Murringo spray dried, JD-S = Defatted Jurien spray dried, M-S = Full fat Murringo spray dried, J-S = Full fat Jurien spray dried.

**Table 1 foods-12-00908-t001:** Proximate composition of protein isolate obtained by different processing techniques; values are the average of three replicates reported as % dry matter (standard deviation excluded for clarity of the table). The same superscript letters in the same column indicate no significant difference, *p* < 0.05.

Treatment Parameters	Sample Details		Protein (%) N × 6.25	Protein (%) N × 5.7	Fat (%)	Carb (%)	Ash (%)	Crude Fibres (%)
Original Flour	*L. albus* cv Murringo	Full Fat (MF)	43.10 ^a^	39.31 ^a^	11.90 ^c^	11.30 ^c^	3.35 ^a^	23.10 ^c^
		Defatted (MFD)	48.34 ^b^	44.09 ^b^	3.80 ^b^	12.30 ^c^	3.46 ^a^	24.90 ^c^
	*L. angustifolius* cv Jurien	Full Fat (JF)	41.00 ^a^	37.39 ^a^	5.83 ^b^	9.20 ^b^	2.57 ^a^	33.20 ^d^
		Defatted (JFD)	44.74 ^a^	40.80 ^a^	1.50 ^a^	9.80 ^b^	2.58 ^a^	34.10 ^d^
Pasteurized Freeze Dried	*L. albus* cv Murringo	Full Fat (M-PF)	83.24 ^c^	75.91 ^c^	17.04 ^d^	<0.50 ^a^	2.64 ^a^	9.65 ^b^
		Defatted (MD-PF)	94.19 ^e^	85.91 ^e^	4.94 ^b^	<0.50 ^a^	2.94 ^a^	8.44 ^a^
	*L. angustifolius* cv Jurien	Full Fat (J-PF)	83.81 ^c^	76.43 ^c^	12.56 ^c^	<0.50 ^a^	2.95 ^a^	8.44 ^a^
		Defatted (JD-PF)	94.35 ^e^	86.04 ^e^	3.71 ^b^	<0.50^a^	3.00 ^a^	7.62 ^a^
Spray Dried	*L. albus* cv Murringo	Full Fat (M-S)	81.41 ^c^	74.25 ^c^	14.98 ^c,d^	<0.50 ^a^	2.57 ^a^	10.68 ^b^
		Defatted (MD-S)	92.20 ^d,e^	84.0 ^d,e^	3.24 ^a^	<0.50 ^a^	2.98 ^a^	7.41 ^a^
	*L. angustifolius* cv Jurien	Full Fat (J-S)	89.48 ^d^	81.60 ^d^	10.60 ^b^	<0.50 ^a^	2.92 ^a^	9.54 ^b^
		Defatted (JD-S)	94.59 ^e^	86.27 ^e^	3.71 ^a^	<0.50	2.99 ^a^	7.29 ^a^
Direct Freeze Dried	*L. albus* cv Murringo	Full Fat (M-F)	87.21 ^d^	79.54 ^d^	12.68 ^b^	<0.5	2.37 ^a^	7.61 ^a^
		Defatted (MD-F)	92.71 ^e^	84.55 ^e^	3.65 ^a^	<0.6	2.48 ^a^	9.32 ^b^
	*L. angustifolius* cv Jurien	Full Fat (J-F)	91.64 ^d,e^	83.58 ^d,e^	11.34 ^b^	<0.7	2.47 ^a^	7.28 ^a^
		Defatted (JD-F)	94.56 ^e^	86.24 ^e^	2.98 ^a^	<0.8	2.78 ^a^	8.00 ^a^

**Table 2 foods-12-00908-t002:** Amino acid composition of lupin protein isolates prepared by different processing techniques. MD-F = Defatted Murringo direct freeze dried, JD-F = Defatted Jurien direct freeze dried, M-F = Full fat Murringo direct freeze dried, J-F = Full fat Jurien direct freeze dried, MD-PF = Defatted Murringo pasteurized and freeze dried, JD-PF = Defatted Jurien pasteurized and freeze dried, M-PF = Full fat Murringo pasteurized and freeze dried, J-PF = Full fat Jurien pasteurized and freeze dried, MD-S = Defatted Murringo spray dried, JD-S = Defatted Jurien spray dried, M-S = Full fat Murringo spray dried, J-S = Full fat Jurien spray dried.

Amino Acids	*L. albus*						*L. angustifolius*					
	Spray Dried		Freeze Dried				Spray Dried		Freeze Dried			
	MD-S	M-S	MD-F	M-F	MD-PF	M-PF	JD-S	J-S	JD-F	J-F	JD-PF	J-PF
Aspartic acid	10.90	11.47	11.60	11.58	10.99	11.02	11.18	11.79	12.33	10.99	11.88	10.94
Arginine	10.80	10.17	10.67	10.73	10.64	10.95	12.35	11.24	13.01	11.42	11.72	11.97
Serine	5.25	4.95	5.48	5.48	4.98	5.01	5.03	4.71	5.47	4.98	4.92	4.95
Glutamic acid	23.41	25.14	25.30	25.10	25.24	24.44	23.86	25.51	27.16	24.10	25.64	23.70
Proline	4.30	3.77	4.41	4.35	3.97	4.03	4.29	3.70	4.57	4.01	3.78	3.98
Glycine	3.86	2.97	3.57	3.62	2.96	2.74	4.03	3.17	3.86	2.99	2.73	3.27
Alanine	3.01	2.85	2.78	2.80	2.81	2.62	2.96	2.99	2.24	2.82	2.49	3.06
**Total non-essential**	**67.62**	**68.41**	**69.70**	**69.60**	**68.22**	**67.62**	**68.60**	**68.77**	**72.56**	**67.41**	**68.61**	**67.16**
Cysteine	0.97	1.36	1.04	1.05	1.18	1.22	0.86	0.94	0.84	1.09	0.89	0.93
Methionine *	0.53	0.37	0.56	0.57	0.33	0.36	0.38	0.40	0.30	0.34	0.27	0.32
**Total sulphur containing**	**1.50**	**1.73**	**0.60**	**0.62**	**1.51**	**1.59**	**1.24**	**1.34**	**0.34**	**1.43**	**1.16**	**1.25**
Tryptophan *	0.54	0.41	0.59	0.60	0.45	0.40	0.62	0.57	0.48	0.47	0.33	0.55
Tyrosine	5.11	5.73	4.85	4.90	5.47	5.59	4.03	4.71	3.09	5.02	4.55	4.35
Phenylalanine *	4.28	4.47	3.76	3.85	4.50	4.61	4.45	4.68	3.21	4.76	4.72	4.93
**Total aromatic**	**9.92**	**10.61**	**9.20**	**9.35**	**10.42**	**10.60**	**9.11**	**9.96**	**6.78**	**10.24**	**9.61**	**9.83**
Threonine *	3.39	3.32	3.13	3.18	3.26	3.36	3.02	2.97	2.20	3.28	3.07	3.19
Lysine *	4.46	4.30	3.75	3.84	4.27	4.34	4.32	4.17	3.03	4.41	4.31	4.49
Valine *	4.18	3.67	3.87	3.79	3.65	3.74	3.93	3.49	3.76	3.74	3.60	3.75
Histidine *	1.99	1.90	1.80	1.89	1.90	1.91	2.31	2.20	1.64	2.06	2.20	2.24
Isoleucine *	5.19	5.65	4.75	4.69	5.75	5.84	5.05	5.59	4.91	5.84	5.69	5.84
Leucine *	7.83	7.48	8.10	7.99	7.67	7.82	7.31	7.15	7.89	7.69	7.19	7.53
**Total Basic**	17.25	16.38	16.22	16.46	16.80	17.20	18.98	17.62	17.68	17.89	18.24	18.70
**Total essential**	**32.38**	**31.59**	**30.30**	**30.40**	**31.78**	**32.38**	**31.40**	**31.23**	**27.44**	**32.59**	**31.39**	**32.84**
**Total polar**	**66.29**	**68.34**	**67.61**	**67.75**	**67.92**	**67.84**	**66.95**	**68.25**	**68.77**	**67.35**	**69.20**	**66.77**
**Total non-polar**	**33.71**	**31.66**	**32.39**	**32.25**	**32.08**	**32.16**	**33.05**	**31.75**	**31.23**	**32.65**	**30.80**	**33.23**

Aspartic acid + asparagine; Glutamic acid + glutamine; * Essential.

## Data Availability

The data is included in the article and Appendix A.

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
