# Peer review of "Structural and Thermal Characterization of Protein Isolates from Australian Lupin Varieties as Affected by Processing Conditions"

_foods, 2023, doi:10.3390/foods12050908_

Round 1

Reviewer 1 Report

1Line 107 In the flowchart, collect all the precipitate and adjust to pH 7.2, why not 7.0? Please explain.

2In the flowchart, the pasteurization condition is set to 75+3 , 5min. Is the sterilization temperature and sterilization time here based on the sterilization conditions of what product? Please give the basis for your conditions. Also, please describe how this sterilization condition relates to the product or application that your subject lupinmay develop.

3Line 252-255in Table 1. You give proximate composition of protein isolate obtained by different processing techniques. However, I think there should be another sample, which is the raw seedlupin, and its protein content and other components can be determined, so that it can be used as a control and compare the impact of your different processing techniques on the protein content.

4Line 226-251  3.1. Protein extraction and proximate analysis For different processing methods, please explain further why the protein content is different. In addition, the literature comparing the conversion factors of N 6.25 and 5.27 can be properly analyzed.

5Line377 in Table 2, Amino acid composition has not been statistically analyzed, and it is not possible to compare the effects of different processing on amino acids, please add.

6Line404-411CD-derived secondary structure of soluble lupin protein isolates is presented in figure4b. Please explain why the secondary structures of soluble proteins in different processing treatments are very different in the CD spectral analysis results and added in the manuscript.

7Line424-436Although the pasteurization temperature is lower than the thermal denaturation temperature, the spray drying temperature is higher than the thermal denaturation temperature, thus allowing the authors to analyze the variation of the sample data with respect to this spray drying. Please add to the article.

Reviewer 2 Report

Totally speaking, this research article regarding the proteins from the full and defatted flours of Australian lupin varieties that were prepared using alkaline extraction and iso-electric precipitation, is very well designed. The current study's goal was to elucidate the varietal and processing induces effects on the molecular and secondary protein's structure. The secondary structure was characterized by Fourier-transform infrared and circular dichroism spectroscopy. Meanwhile, the thermal characterization and amino acid profile of protein isolate were summarized. In conclusion, the authors emphasized that the commercial processing conditions (pasteurization, spray dryer parameters, and liophilization parameters) did not have a profound effect on the various structural properties of lupin protein isolates and that these properties were mainly determined by varietal differences. This study was suitable for Foods journal, and proposed special issue. However, there are a few points that require clarification before acceptance for publication.

(1) Lines 6 to 12: Please include the matching e-mail addresses of the co-authors listed above, as well as their initials in parenthesis.

(2) Abstract: According to the existing literature, this section of the manuscript requires a description of the scientific background linked to the separated proteins. Also, discuss the work's purpose in greater detail before moving on to the methods utilized and the results.

(3) Keywords: Because this study is about protein secondary structure and process parameters that influence the structure of proteins, strengthen the key ones with, say, circular analysis instead of size exclusion chromatography, and also add the process parameters of drying and pasteurization.

(4) Introduction: Please finish the introduction with particular statistics on protein consumer needs, annual plant protein production levels, and food sector demand for them. Next, isolate the procedures for obtaining plant proteins as separate procedures and compare the extraction used in this paper with the literature isolation procedures.

(5) Line 139: ''TGX precast gels and Tris/Glycine/SDS buffer'' Please, specify the manufacturer.

(6) Line 151: Which type of 150 mM salt was used? Sodium-chloride or sodium-sulfate?

(7) Line 153: Please specify the procedure for constructing the calibration curve and the calculation utilized for the determination of molecular weight.

(8) Line 167: Were amino acid standards used for the construction of standard solutions? Mention which and briefly describe the procedure for calculating the amino acid content in samples in accordance with the calibration curve.

The total amino acid content was examined; what happened with free? Please, specify.

(9) It is advised that the authors recheck the main text during the revision to make this manuscript more readable.

(10) References: Authors are asked to arrange the list of cited papers in the references section according to the journal's instructions; each paper, book, or chapter must also contain a number.

Reviewer 3 Report

The authors produced protein isolates from two varieties of lupine and studied how defatting and different drying regimes affected the protein structure. The article is well written, and it was a pleasure to read. The experimental design and the analysis methods were appropriate and support the conclusions. I did not find any major issues; only small corrections are needed:

The first paragraph/sentence of Introduction should give clear indication for the topic of the article. At the moment, it reads as if this article is about sustainability, but instead it is about properties of lupin protein isolates.

The sample abbreviation scheme was confusing, and I found myself lost until the very end of the article. There are many samples, but the naming pattern is inconsistent, and F means different things. Furthermore, one would naively expect more letters would mean more processing, but instead JIF is less processed than JI and MIFS is not made from MIF. Letter “I” appears everywhere and thus redundant. Consider this scheme: M, MD (M and M Defatted) and then adding PL, L, S for Pasteurized Lyophilized (to avoid association with F/fat), Lyophilized, and Spray dried. So, MI becomes MD-PL. A bit longer, but it is also consistent and easier to remember.

Method 2.2.  Did you follow any previously published protocol? Add some more information about the extraction process. Line 100: Did you constantly add more NaOH or just once? Line 101: by “fraction” you mean “supernatant”? Line 102: Any reason for overnight precipitation? Line 103: Resuspended in what amount of water? Line 104: pH maintained by what? Line 105: Describe pasteurization in more detail. Specify spray drying equipment and add here the in/out temperatures. What was the dry matter content before spray drying (approximately)?

Line 137. “Laemmle” -> “Laemmli”.

Line 443, also 450. “A principal component analysis and biplot” -> “A PCA loading plot and biplot”. Biplot is not a different type of PCA.

Line 480. “Analysis of revealed” -> “Analysis revealed”

Line 490-493. Some extra text left.

Round 2

Reviewer 1 Report

1Line360-363 Please explain why the secondary structures of soluble proteins in different processing treatments are very different in the CD spectral analysis results and added in the manuscriptAdd some additional explanations as appropriate and attach references.

2Please emphasizing how your results can contribute to food scienceadd a few more word in conclusions.

Author Response

Thanks 
